# The Clinical Effect of a Propolis and Mangosteen Extract Complex in Subjects with Gingivitis: A Randomized, Double-Blind, and Placebo-Controlled Clinical Trial

**DOI:** 10.3390/nu16173000

**Published:** 2024-09-05

**Authors:** Jae-Suk Jung, Geum-Hee Choi, Heelim Lee, Youngkyung Ko, Suk Ji

**Affiliations:** 1Department of Periodontology, Institute of Oral Health Science, Ajou University School of Medicine, Suwon 16499, Republic of Korea; jsjung84@ajou.ac.kr (J.-S.J.); epni33@naver.com (G.-H.C.); heelimlee@gmail.com (H.L.); 2Department of Dentistry, College of Medicine, The Catholic University of Korea, Seoul 06591, Republic of Korea

**Keywords:** propolis, mangosteen, gingivitis, anti-inflammatory effect

## Abstract

This study investigated the efficacy and safety of a propolis–mangosteen extract complex (PMEC) on gingival health in patients with gingivitis and incipient periodontitis. A multicentered, randomized, double-blind, placebo-controlled trial involving 104 subjects receiving either PMEC or placebo for eight weeks was conducted. The primary focus was on the changes in inflammatory biomarkers from gingival crevicular fluid (GCF), with clinical parameters as secondary outcomes. The results revealed that the PMEC group showed a significantly reduced expression of all measured GCF biomarkers compared to the placebo group (*p* < 0.0001) at 8 weeks, including substantial reductions in IL-1β, PGE2, MMP-8, and MMP-9 levels compared to the baseline. While clinical parameters trended towards improvement in both groups, the intergroup differences were not statistically significant. No significant adverse events were reported, indicating a favorable safety profile. These findings suggest that PMEC consumption can attenuate gingival inflammation and mitigate periodontal tissue destruction by modulating key inflammatory mediators in gingival tissue. Although PMEC shows promise as a potential adjunctive therapy for supporting gingival health, the discrepancy between biomarker improvements and clinical outcomes warrants further investigation to fully elucidate its therapeutic potential in periodontal health management.

## 1. Introduction

Periodontal disease is a chronic inflammatory condition characterized by the progressive destruction of tooth-supporting tissues, resulting from dysregulated host immune responses to pathogenic biofilms [1]. The pathogenesis involves complex interactions between oral microbiota and host immune cells, leading to the production of pro-inflammatory cytokines, prostaglandins, and matrix metalloproteinases (MMPs), which contribute to soft tissue inflammation and alveolar bone resorption [2].

The diagnostic criteria for periodontal disease typically include the gingival index (GI), bleeding on probing (BOP), the periodontal pocket depth (PD), gingival recession (GR), the clinical attachment level (CAL), and the plaque index (PI) [3]. Clinical periodontal examination is mandatory and provides basic information for patients in the diagnosis of periodontal disease; however, these clinical measures provide information on past periodontal tissue destruction and they are inadequate for determining current periodontitis activity or prognosis [3]. Moreover, they exhibit limitations such as low sensitivity and high false-positive rates [4,5]. Consequently, research efforts have focused on identifying periodontal disease biomarkers capable of predicting disease activity or progression [3,6]. Gingival crevicular fluid (GCF) has emerged as a rich source of potential biomarkers, offI ering a non-invasive means to assess the local inflammatory status of periodontal tissues [7]. GCF contains bacterial metabolites and inflammatory exudates produced by periodontal tissues, including capillaries, in response to bacterial challenge during disease progression. Its non-invasive collection method further enhances its utility in research [1,3,6,8,9]. Several molecules in GCF have been investigated as potential biomarkers, including pro-inflammatory cytokines, proteolytic enzymes, and inflammatory mediators [10,11]. Interleukin-1β (IL-1β) is a key pro-inflammatory cytokine that plays a crucial role in the pathogenesis of periodontal disease [2,7]. Prostaglandin E2 (PGE2), an arachidonic acid metabolite, is another important mediator of inflammation and bone resorption in periodontal tissues. Studies have shown that the PGE2 levels in GCF correlate with clinical parameters of periodontal disease severity [10]. Matrix metalloproteinases, particularly MMP-8 and MMP-9, have gained significant attention as potential biomarkers for periodontal disease. MMP-8, also known as neutrophil collagenase, is a key enzyme involved in the degradation of type I collagen, the primary structural protein in periodontal tissues. Elevated levels of MMP-8 in GCF have been consistently associated with periodontal disease activity and progression [12]. Similarly, MMP-9, or gelatinase B, has been implicated in the breakdown of type IV collagen and other extracellular matrix components. Both MMP-8 and MMP-9 levels in GCF have shown promise as diagnostic and prognostic markers for periodontal disease [11]. The detection of periodontal disease biomarkers in GCF holds significant potential for assessing, diagnosing, and prognosticating periodontal disease activity [11,12,13,14].

The association between periodontal disease and systemic conditions such as diabetes mellitus and cardiovascular disease underscores the importance of effective prevention and early intervention strategies [15,16]. While antibiotics have been traditionally employed in periodontal therapy, their long-term use is limited by the emergence of resistant bacterial strains and the risk of opportunistic infections. Consequently, there is growing interest in developing safe and efficacious natural compounds with anti-inflammatory and antimicrobial properties [17,18].

*Garcinia mangostana* L. (mangosteen) and propolis have gained attention for their potential therapeutic properties in periodontal health. Mangosteen contains bioactive compounds, including xanthones and flavonoids, with demonstrated anti-inflammatory and antimicrobial effects [19,20]. Propolis, a resinous substance produced by honeybees, contains various bioactive compounds such as flavonoids and phenolic acids with antioxidant and immunomodulatory properties [21,22,23]. It exhibited antimicrobial activity against Gram-positive and Gram-negative bacteria, as well as fungi, and demonstrated antioxidant activity by scavenging free radicals, inhibiting hemolysis, and preventing lipid peroxidation in human erythrocytes incubated with an oxidizing agent [23].

Previous in vitro and animal studies have demonstrated the potential synergistic effects of propolis and mangosteen extracts in modulating inflammatory responses and promoting osteogenic activity [24,25,26]. However, clinical evidence supporting the efficacy of this combination in managing periodontal health remains limited. Therefore, the present study aimed to evaluate the efficacy and safety of a propolis–mangosteen extract complex (PMEC) on gingival health in subjects with gingivitis and incipient periodontitis. We hypothesized that PMEC supplementation would modulate the expression of key inflammatory mediators (IL-1β, PGE2, MMP-8, and MMP-9) in GCF and improve the clinical parameters of periodontal health. By focusing on both biomarker analysis and clinical assessments, this study seeks to provide a comprehensive evaluation of the potential benefits of PMEC in periodontal health management.

## 2. Materials and Methods

### 2.1. Study Design and Participants

This study was designed as a multicenter, double-blind, placebo-controlled, simple randomized clinical trial and was registered with the Clinical Research Information Service of the National Research Institute of Health in the Republic of Korea (KCT0008940). All procedures adhered to the Declaration of Helsinki and were approved by the Institutional Review Board of the Ajou University Medical Center (AJIRB-MED-FOD-21-607) and the Catholic University of Korea, Seoul St. Mary’s Hospital, College of Medicine (KC22HSDE0008). A total of 104 subjects were enrolled within the Department of Periodontics at Ajou University Dental Hospital and Department of Periodontics, Seoul St. Mary’s Hospital, Catholic University of Korea. Prior to enrollment, all participants were informed of the nature of this study, and informed consent was obtained. 

### 2.2. Intervention and Monitoring

Participants were randomly assigned to the following groups. The control group received the same placebo capsules without the PMEC as the test group. The dosage and composition of the PMEC were determined based on previous preclinical studies. The extracted material was formulated in a 2:68 ratio (mangosteen–propolis) by weight. The test group took 2 capsules containing 194 mg of PMEC once daily for 8 weeks (56 days). When converted for a 70 kg adult human, the dose was 388 mg of PMEC. Details of the ingredients of the test and placebo capsules are shown in Table 1 (Appendix A).

### 2.3. Selection Criteria: Inclusion/Exclusion Criteria

This study was conducted in subjects with gingivitis and incipient periodontitis. The inclusion criteria were (1) adult men and women aged over 20 years and under 70 years with mild gum disease, (2) at least 20 natural teeth, (3) at least 10% BOP sites in all teeth, and (4) at least one tooth with a PD of at least 3 mm but not more than 5 mm and a GI of at least 2 points. Exclusion criteria were (1) clinically significant cardiovascular, immune system, or infectious disease; (2) psychiatric disorders such as schizophrenia, depression, or substance addiction; (3) cancer within the past 5 years; (4) a history of bleeding disorders or conditions or taking antiplatelet or anticoagulant medications to prevent them; (5) significant pathologic findings in the soft tissues of the oral cavity; (6) pregnancy or lactation; (7) smoking; (8) scaling treatment within 3 months prior to screening or periodontal treatment within 6 months; (9) medications that may affect periodontal status within 1 month prior to screening (Phenytoin, CCBs, Cyclosporine, Coumadin, NSAIDs, Aspirin, etc.) for more than 5 consecutive days; (10) antibiotics or periodontal supportive therapy within 1 month prior to screening; (11) dietary supplements related to gum health within 1 month prior to screening; (12) uncontrolled hypertension (systolic blood pressure greater than or equal to 160 mmHg or diastolic blood pressure greater than or equal to 100 mmHg, measured after 10 min of patient rest); (13) uncontrolled diabetes; (14) creatinine greater than or equal to 2 times the upper limit of site normal; (15) participation in another clinical trial; or (16) other reasons deemed unsuitable for inclusion by the clinician. 

### 2.4. Clinical Parameters and Biomarkers of Gingival Crevicular Fluid (GCF)

The clinical parameters used to evaluate the validity of this study were GI, PD, BOP, GR, CAL, and PI. GI was measured a modified version of the method described by Löe and Silness [27]. Teeth were divided into buccal and lingual surfaces for both the maxilla and mandible. Each surface was rated on a scale of 0 to 3, where 0 indicates healthy gingiva and 3 indicates severe inflammation. The mean GI for an individual was calculated by averaging the scores of all examined teeth. PD was measured by assessing the distance from the gingival margin to the periodontal pocket along the tooth surface. BOP was determined positive if bleeding was observed within 30 s following probe placement, and the BOP percentage (BOP%) was calculated as the ratio of BOP-positive sites to the total number of sites examined, expressed as a percentage. GR was measured as the distance from the cementoenamel junction (CEJ) to the gingival margin. CAL measurements were performed based on the periodontal tissues at the CEJ to the site of periodontal tissue attachment and were calculated as the sum of PD and GR. PD, BOP, GR, and CAL were measured at six sites [proximal, mesial, central, and distal on the buccal surface/lingual surface] for each tooth. PI was categorized into buccal–lingual for maxillary and mandibular teeth and scored on a 0–5 scale [28]. All clinical parameters were measured by the same examiner at visit 1 (screening) or 2 (baseline measurement) and visit 3 (after taking PMEC or placebo for 4 weeks) and 4 (after taking PMEC or placebo for 8 weeks).

Gingival crevicular fluid (GCF) samples were collected to analyze specific biomarkers: IL-1β, PGE2, MMP-8, and MMP-9. Sample collection was performed at visits 1 or 2 and repeated by the same investigator at visits 3 and 4 to ensure consistency. GCF samples were obtained using Periopaper strips (ProFlow Inc., Amityville, NY, USA) inserted for 60 s into the gingival sulcus of the tooth exhibiting the deepest periodontal pocket, sampling four sites per tooth: mesial buccal, mesial lingual, distal buccal, and distal lingual surfaces. Following collection, samples were immediately frozen to preserve biomarker integrity. Analysis was conducted at the Department of Periodontology, Institute of Oral Health Science, Ajou University School of Medicine, using standardized protocols for each biomarker. The concentrations of IL-1β, PGE2, MMP-8, and MMP-9 in gingival crevicular fluid (GCF) were quantified using enzyme-linked immunosorbent assay (ELISA) kits (R&D Systems, Minneapolis, USA; IL-1β: DY201; PGE2: PKGE004B; MMP-8: DY908; MMP-9: DY911) following the manufacturer’s protocols. Upon completion of the analysis, all specimens were appropriately discarded in accordance with institutional biosafety guidelines.

### 2.5. Safety Analysis Method

The safety evaluation was performed using a safety set analysis as the primary analysis and included 51 test subjects and 53 control subjects who were randomized to the human clinical trial and consumed the human clinical trial food at least once. The type, the incidence, and the severity of the adverse events and their association with the investigational product were evaluated. In addition, the results of clinicopathological examinations (hematology/hematochemistry, urinalysis), vital signs (blood pressure, pulse), and anthropometric measurements (body weight) were analyzed at screening and 8 weeks.

### 2.6. Data Set Characterization

The data collected in this study were categorized into the safety set, full analysis (FA) set, and per protocol (PP) set. The safety set included all participants who consumed at least one dose of the test product. Based on the intention-to-treat (ITT) protocol, the FA set included all participants who received at least one dose of the test product, attended efficacy assessments at weeks 4 and 8, and met the inclusion/exclusion criteria. The PP set included only participants who completed this study in compliance with the inclusion/exclusion criteria. The PP set was mainly used to assess efficacy, with additional FA set analyses. The safety set was used for safety analysis only. 

### 2.7. Statistical Analysis

Statistical analyses were performed using SAS^®^ (Version 9.4, SAS Institute, Cary, NC, USA). Data on efficacy, demographic and nutritional analysis, and safety were subjected to two-tailed tests with a significance level of 0.05. Data are presented as mean and standard deviation, and statistical significance of differences between groups was defined as *p* < 0.05. Comparisons between groups were analyzed using a normality test (Shapiro–Wilk) at a *p*-value of 0.05, followed by a two-sample *t*-test if both test and control groups met normality and a Wilcoxon rank sum test if either group failed to meet normality. In the two-sample *t*-test, the Last Observation Carried Forward (LOCF) method was utilized to address missing data within the FA set. Specially, for missing data at the 8-week time point, the corresponding 4-week data were inputted for analysis. However, missing data at the 4-week time point were not substituted with baseline measurements. Additionally, a repeated-measures 2-way ANOVA was performed and analyzed using the results of repeated measurements taken before, 4 weeks after, and 8 weeks after consuming the test food. Subset (group) analysis was performed by categorizing the initial characteristics of the human clinical trial subjects before randomization or before treatment initiation to allow for further analysis.

## 3. Results

### 3.1. Participant Flow and Baseline Characteristics

A total of 118 participants were initially assessed for eligibility, with 14 failing screening. The remaining 104 participants were randomized into test (*n* = 51) and control (*n* = 53) groups, forming the safety set. Subsequently, participants progressed through full analysis (FA) and per protocol (PP) sets. The FA set comprised 97 participants (test group: *n* = 48; control group: *n* = 49) after excluding 3 from the test group (2 for inclusion/exclusion criteria deviations, 1 lost to follow-up) and 4 from the control group (all for inclusion/exclusion criteria deviations). The PP set was further refined to 94 participants (test group: *n* = 46; control group: *n* = 48) following additional exclusions (test group: 1 inclusion/exclusion criteria deviation, 1 lost to follow-up; control group: 1 inclusion criteria failure). Figure 1 illustrates this participant flow. The baseline demographics and clinical characteristics showed no statistically significant differences between groups (Table 2). This study design allowed for comprehensive analysis at multiple levels of participant inclusion, ensuring robust evaluation of the intervention’s effects.

### 3.2. Clinical Parameters

The analysis of clinical parameters revealed nuanced trends across different measures. The GI in both test and control groups demonstrated a statistically significant within-group decrease at 8 weeks compared to the baseline in both the PP and FA sets (*p* < 0.05), although the between-group differences were not significant (Table 3 and Table 4). BOP (%) significantly decreased in the control group (PP set, Table 3) and in both groups (FA set, Table 4) from the baseline to 8 weeks. The PD exhibited a decreasing trend in the test group and significant decreases in the control group at both 4 and 8 weeks (Table 3 and Table 4). GR in the test group significantly increased from the baseline to 4 weeks in both sets (*p* < 0.05, Table 3 and Table 4). The CAL showed a significant decrease in the control group from 4 weeks onward in both sets (*p* < 0.05, Table 3 and Table 4). The PI demonstrated significant reductions in the test group at both 4 and 8 weeks, while the control group showed a significant reduction only at 8 weeks. Importantly, between-group comparisons for all clinical parameters did not reach statistical significance at any time point (all *p* > 0.05, Table 3 and Table 4 and Figure 2).

### 3.3. Biomarkers of Gingival Crevicular Fluid (GCF)

The analysis of GCF biomarkers revealed significant changes in the expression of key inflammatory mediators over the course of this study. In the test group, the expression of IL-1β, PGE2, MMP-8, and MMP-9 showed marked decreases from the baseline to 4 and 8 weeks in both the PP and FA sets (*p* < 0.0001 for all, except IL-1β (*p* ≤ 0.005 at 4 weeks), Table 5 and Table 6). Specifically, at 8 weeks, IL-1β decreased to 42.1 ± 9.7%, PGE2 to 48.9 ± 8.6%, MMP-8 to 53.7 ± 8.1%, and MMP-9 to 45.6 ± 7.3% of baseline levels in the PP set (Figure 3a). Conversely, the control group exhibited increased trends in all biomarkers, with IL-1β reaching 138.2 ± 18.6%, PGE2 129.6 ± 16.2%, MMP-8 124.5 ± 15.7%, and MMP-9 133.8 ± 17.5% at 8 weeks (Figure 3). Between-group comparisons revealed statistically significant differences in all four biomarkers at 8 weeks (*p* < 0.001 for all comparisons, Table 5 and Table 6 and Figure 3). The test group experienced a more rapid decrease in biomarker levels during the first 4 weeks (average rate of 7.9% per week) compared to the subsequent 4 weeks (6.5% per week). Strong positive correlations were observed between the reductions in different biomarkers (Pearson’s r: 0.72–0.89, *p* < 0.001), indicating a coordinated modulation of the inflammatory response. These results provide strong evidence for the efficacy of PMEC in modulating key inflammatory mediators in GCF, with consistent and significant reductions observed across all measured biomarkers, contrasting with the increases seen in the control group.

### 3.4. Safety Analysis

Safety evaluations, including clinicopathological examinations (hematology, biochemistry, urinalysis), vital signs (pulse, blood pressure), and anthropometric measurements (body weight), showed no statistically significant differences between the test and control groups after 8 weeks of treatment (Table 7). No serious adverse events were reported during the study period.

## 4. Discussion

This randomized, double-blind, placebo-controlled clinical trial evaluated the efficacy of a propolis–mangosteen extract complex (PMEC) on gingival health in subjects with mild to moderate gingivitis. Our findings demonstrate that PMEC supplementation significantly modulated the expression of key inflammatory mediators in gingival crevicular fluid (GCF), although clinical parameters showed only modest improvements. 

The most striking outcome of this study was the significant reduction in GCF biomarkers (IL-1β, PGE2, MMP-8, and MMP-9) observed in the PMEC group compared to the placebo group. These biomarkers are well-established indicators of periodontal inflammation and tissue destruction [1,9,11,12,29]. The consistent downregulation of these mediators suggests that PMEC may exert a potent anti-inflammatory effect in the periodontal microenvironment. The observed decrease in PGE2 levels (51.1% reduction at 8 weeks) is particularly noteworthy. PGEs, derivatives of arachidonic acid metabolism, are prevalent at inflammatory sites [30]. PGE2, in particular, plays a crucial role in osteoclastogenesis and bone resorption. Its expression is elevated in both periodontal tissue and gingival crevicular fluid as periodontitis advances [10,31]. Notably, PGE2 is theorized to be a primary mediator of the inflammatory and destructive changes observed in periodontal disease, including gingival erythema, edema, collagen degradation, and alveolar bone loss [1].

IL-1β, a primary mediator of the inflammatory response, plays a crucial role in periodontal pathogenesis [2]. The significant reduction in IL-1β levels (57.9% decrease at 8 weeks) in the PMEC group may indicate a dampening of the pro-inflammatory cascade. In response to oral pathogens, neutrophils and macrophages in periodontal tissues release cytokines such as IL-1β, which amplify the adaptive immune response aimed at bacterial elimination. Concurrently, neutrophil-derived MMPs, particularly MMP-8 and MMP-9, contribute to the degradation of periodontal soft and hard tissues by breaking down their primary matrix protein, collagen [9,32]. This finding aligns with previous studies demonstrating the anti-inflammatory properties of propolis or mangosteen extracts [33,34]. The significant decreases in MMP-8 (46.3% reduction) and MMP-9 (54.4% reduction) levels are particularly relevant to periodontal health. These matrix metalloproteinases are primary enzymes responsible for extracellular matrix degradation in periodontal tissues, with MMP-8 specifically targeting type I collagen, the most abundant protein in the periodontal ligament [35]. Their downregulation implies that PMEC may help preserve periodontal tissue integrity by reducing proteolytic activity. Previous studies have shown that both propolis and mangosteen components can inhibit MMPs’ expression and activity [25,36], supporting our findings. These results demonstrate that the consumption of PMEC may confer beneficial effects on periodontal tissue health through its anti-inflammatory properties, as evidenced by the observed reduction in key periodontal disease biomarkers, including IL-1β, PGE2, MMP-8, and MMP-9 [Figure 4].

While both the PMEC and placebo groups demonstrated improvements in various periodontal health clinical parameters, there were no significant intergroup differences. The reason for this discrepancy is not clear, but the oral hygiene control ability of subjects is one of the potential factors, as the oral administration of PMEC cannot control the oral bacteria on the dental surface (Figure 4). At baseline, the GI and BOP values of the test group were higher than those of the control group (Table 3), which suggests that the test group participants’ oral hygiene control ability was less effective than that of the control group in the past, before participating in this clinical trial. These results suggest that all participants in this study seemed to have paid more attention to oral hygiene during the test period, but the participants in the control group seemed to have slightly better oral hygiene skills. This increased attention to oral hygiene by all participants could be attributed to the Hawthorne effect, a phenomenon where individuals modify their behavior in response to their awareness of being observed [37]. In this case, the participants’ knowledge of being part of a clinical trial may have influenced their oral hygiene practices, potentially masking any direct effects of the PMEC treatment. Additionally, to maximize the effects of PMEC, it is thought that the effects will increase if oral hygiene is thoroughly managed.

The clinical parameters used in this study have been widely used for the optical diagnosis of periodontal diseases. However, these clinical parameters may miss subclinical inflammation or the early stages of periodontal disease. In the early or asymptomatic stages of inflammation, there may be no noticeable clinical signs such as bleeding or gingival color changes, making it difficult to detect these conditions with conventional methods [38] (p. 33). The main clinical parameters that show the degree or presence of gingival inflammation are the GI and BOP. BOP is affected by the pressure applied during probing, which can cause bleeding that may not be related to actual inflammation, resulting in very low sensitivity and high false positives [4,5]. In the GI, the subjectivity of clinical periodontal assessment has been reported to be the least reproducible, with the lowest reproducibility in both intra-examiner and inter-examiner comparisons of the five clinical parameters compared [39]. Evidence of periodontal tissue attachment loss can be assessed by the CAL, which is calculated as the sum of the PD and GR. However, these clinical periodontal parameters provide information about the past and require a significant amount of damage to provide information about the extent of periodontal destruction [38] (p. 33). It has also been reported that the results of periodontal probing are limited, influenced by several factors such as the design of the probe tip, the pressure exerted by the probe, the degree of inflammation of the soft tissue affecting resistance, and the angle of the probe [40]. This results in low intra- and inter-examiner reproducibility; in fact, for manual probes, the mean intra-examiner standard deviation of repeated site probing depth measurements ranged from 0.52 to 0.89 mm [41]. Additionally, the interval of the scale on the periodontal probe is 1 mm. This means that the smallest change we can reliably measure is 1 mm, with a potential error of about 0.5 mm in either direction. In our study, the changes in the average value of the PD between the baseline and 8 weeks were 0.04 and 0.07 in the test and control groups, respectively. The average change value of the PD is much less than the potential error of the periodontal probe, and the observed alterations in the PD values for both the test and control groups might be negligible. These limitations of clinical parameters have led to the validation of various biomarkers to show the extent of periodontitis progression in gingival crevicular fluid (GCF) or saliva, and several inflammatory mediators have been shown to reflect the extent of the disease [7]. Several studies have shown that MMP-8 and IL-1β are the most reliable markers of persistent periodontitis [42,43], and the Helsinki group has developed point-of-care tests (PerioSafe^®^, ImplantSafe^®^) to assess salivary MMP-8 levels. Other studies have identified an association between periodontal disease and increased PGE2 [44] and MMP-9 [45] in GCF.

While PMEC shows promise as a potential adjunctive therapy for supporting gingival health, particularly in early-stage periodontal disease, the discrepancy between biomarker improvements and clinical outcomes warrants further investigation. Future studies with larger cohorts, extended durations, and exploration of optimal dosing regimens are necessary to elucidate the full therapeutic potential of PMEC in periodontal health management. 

In our study, we consider the results from the full analysis (FA) set to be of primary importance. The FA set, which follows the intention-to-treat principle, includes all randomized participants regardless of protocol deviations or withdrawals. This approach preserves the integrity of randomization, reduces potential bias, and provides a more conservative and realistic estimate of the treatment effect. The FA set results better reflect real-world scenarios where perfect adherence to treatment protocols is not always achieved. This aligns with regulatory preferences and offers a more pragmatic view of the intervention’s effectiveness in a general population. While we also present the PP set results for completeness and to provide insights into the treatment’s efficacy under ideal conditions, we prioritize the FA set findings in drawing our main conclusions. The PP set serves as a valuable complement, offering a “best-case scenario” and allowing for sensitivity analysis. By emphasizing the FA set results while also reporting the PP set data, we aim to provide a comprehensive and balanced view of our intervention’s effects, ensuring that our conclusions are both robust and clinically relevant.

## 5. Conclusions

In conclusion, this study provides evidence that PMEC supplementation can significantly modulate key inflammatory mediators in the gingival microenvironment. However, it is important to note that these biochemical changes did not translate into significant clinical improvements in the short term. While the reduction in inflammatory biomarkers suggests potential for PMEC as an adjunctive therapy in periodontal health management, this potential should be viewed cautiously given this study’s limitations, such as its short duration and small sample size. Further studies are needed to validate the potential of PMEC as a natural, holistic approach to periodontal health, employing more rigorous methodologies and extended follow-up periods to conclusively determine its clinical efficacy and long-term benefits in the prevention and management of periodontal disease.

## Figures and Tables

**Figure 1 nutrients-16-03000-f001:**
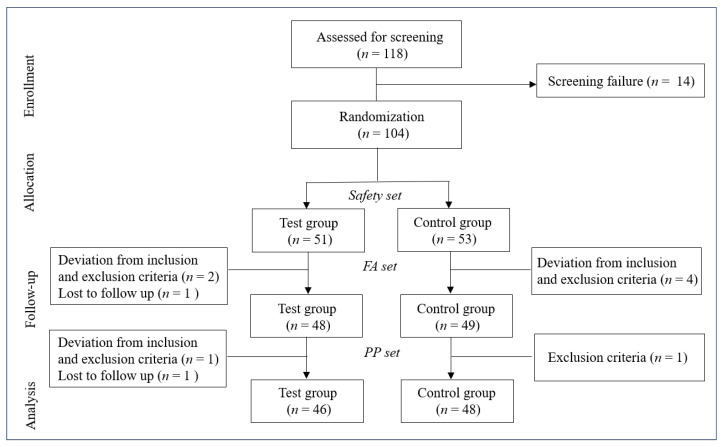
Flowchart depicting the study design. FA, full analysis; PP, per protocol.

**Figure 2 nutrients-16-03000-f002:**
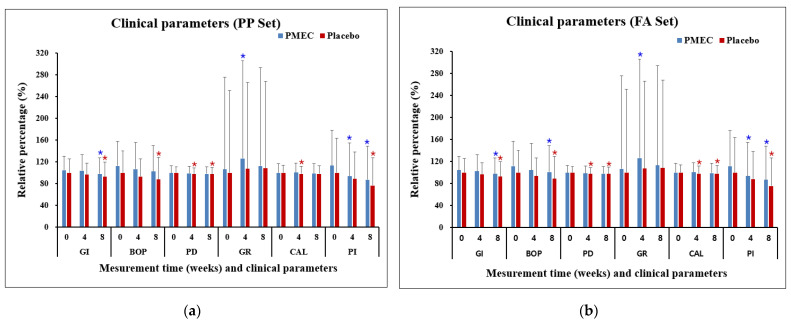
Changes in clinical parameters over time in the PP set (**a**) and the FA set (**b**). Changes are presented as mean percentages ± standard deviation (SD) relative to the baseline of the control group. Each data point represents the average of triplicate measurements. Blue and red asterisks denote statistically significant differences (*p* < 0.05) in the PMEC and control groups, respectively, at weeks 4 and 8 compared to week 0 (baseline). No significant difference was observed between the PMEC and placebo groups at the evaluated time points; consequently, these data are not presented.

**Figure 3 nutrients-16-03000-f003:**
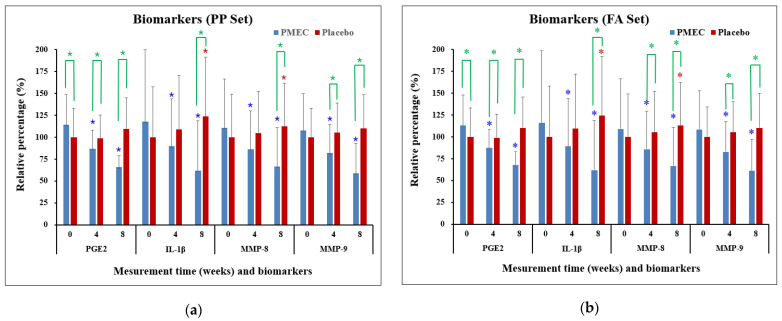
Changes in inflammatory biomarkers in GCF over time in the PP set (**a**) and the FA set (**b**). Changes are presented as mean percentages ± standard deviation (SD) relative to the baseline of the control group. Each data point represents the average of triplicate measurements. Blue and red asterisks denote statistically significant differences (*p* < 0.05) in the PMEC and control groups, respectively, at weeks 4 and 8 compared to week 0 (baseline). Green asterisks denote statistically significant differences (*p* < 0.05) between the PMEC and placebo groups at weeks 0, 4, and 8.

**Figure 4 nutrients-16-03000-f004:**
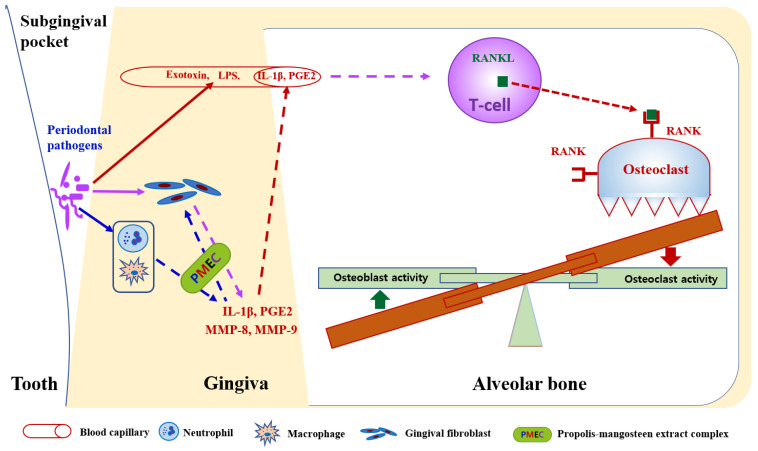
Proposed mechanism of propolis–mangosteen extract complex (PMEC) in mitigating periodontal tissue destruction. PMEC exhibits inhibitory effects on the expression of pro-inflammatory cytokines (IL-1β and PGE2) and metalloproteases (MMP-8 and MMP-9) in oral bacterial-infected periodontal tissue cells, including neutrophils, macrophages, and gingival fibroblasts. This inhibition potentially leads to decreased osteoclastogenesis and enhanced osteogenesis, consequently reducing alveolar bone destruction. However, PMEC does not prevent the direct invasion of periodontal pathogens into gingival tissue cells or the subsequent release of lipopolysaccharides (LPSs) and exotoxins into gingival capillaries. This limitation may explain the discrepancy between clinical parameter outcomes in PMEC treatment and placebo groups, despite significant reductions in gingival crevicular fluid biomarkers (IL-1β, PGE2, MMP-8, and MMP-9) observed in PMEC-treated subjects compared to controls.

**Table 1 nutrients-16-03000-t001:** The details of the constituents of the test and placebo capsules.

	Test		Placebo	
Raw Material	Compounding Ratio (%)	Content (mg)	Compounding Ratio (%)	Content (mg)
Propolis–mangosteen extract complex (PMEC)	41.28	194.00	-	-
Lactose powder	30.00	141.00	62.70	294.69
Microcrystalline cellulose	24.22	113.85	34.27	161.07
Sucrose esters of fatty acids	2.00	9.10	-	-
Caramel color	-	-	2.00	9.40
Magnesium stearate	1.50	7.05	1.00	4.70
Silicon dioxide	1.00	4.70	-	-
Food blue NO.1	-	-	0.03	0.14
Total	100	470	100	470

**Table 2 nutrients-16-03000-t002:** Demographics and clinical characteristics of participants.

Variables	Test Group (*n* = 46)	Control Group (*n* = 48)	*p*-Value ^a^
Age (years), mean ± SD	43.17 ± 11.01	42.02 ± 11.40	0.619 (T)
Gender, *n* (%)			
Male	16 (34.78)	16 (33.33)	0.882 (C)
Female	30 (65.22)	32 (66.67)	
Smoking status, *n* (%)			
No	46 (100.00)	48 (100.00)	-
Yes	0 (0.00)	0 (0.00)	
Physical activity, *n* (%)			
None	8 (17.39)	14 (29.17)	0.536 (C)
1–2 times/week	12 (26.09)	10 (20.83)	
3 times/week	14 (30.43)	14 (29.17)	
4–5 times/week	5 (10.87)	2 (4.17)	
7 times/week	7 (15.22)	8 (16.67)	
Drinking status, *n* (%)			
No	16 (34.78)	16 (33.33)	0.882 (C)
Yes	30 (65.22)	32 (66.67)	
Weight (kg)			
Mean ± SD	64.1 ± 11.8	65.8 ± 12.9	0.498 (W)
Median	61.7	63.7	
Min, Max	48.0, 97.6	46.1, 101.3	
Height (cm)			
Mean ± SD	165.1 ± 7.9	164.6 ± 8.0	0.814 (W)
Median	164.9	164.00	
Min, Max	154.0, 178.0	149.0, 183.0	

^a^ Compared between groups; *p*-value for two-sample *t*-test (T), Chi-square test (C), or Wilcoxon rank sum test (W).

**Table 3 nutrients-16-03000-t003:** Change in clinical parameters by measurement time in the PP set.

Clinical Parameters	Measurement Time	Test (Mean ± SD)	Control (Mean ± SD)	*p*-Value ^b^	*p*-Value ^c^
GI	Baseline	1.70 ± 0.41	1.62 ± 0.42	0.178 (W)	
	4 weeks	1.68 ± 0.48	1.57 ± 0.34	0.067 (W)	
	*p*-value ^a^	0.656	0.189		
	8 weeks	1.59 ± 0.47	1.50 ± 0.44	0.151 (W)	0.864
	*p*-value ^a^	0.026	0.002		
BOP (%)	Baseline	47.26 ± 18.90	41.93 ± 16.69	0.150 (T)	
	4 weeks	44.70 ± 20.43	39.03 ± 13.60	0.131 (W)	
	*p*-value ^a^	0.239	0.077		
	8 weeks	42.98 ± 20.02	36.88 ± 16.95	0.114 (T)	0.955
	*p*-value ^a^	0.057	0.001		
PD	Baseline	2.58 ± 0.35	2.59 ± 0.28	0.649 (W)	
	4 weeks	2.56 ± 0.34	2.53 ± 0.31	0.747 (W)	
	*p*-value ^a^	0.448	0.001		
	8 weeks	2.54 ± 0.33	2.52 ± 0.32	0.987 (W)	0.323
	*p*-value ^a^	0.111	0.004		
GR	Baseline	0.18 ± 0.28	0.17 ± 0.25	0.926 (W)	
	4 weeks	0.21 ± 0.30	0.18 ± 0.27	0.519 (W)	
	*p*-value ^a^	0.002	0.316		
	8 weeks	0.19 ± 0.30	0.18 ± 0.27	0.769 (W)	0.318
	*p*-value ^a^	0.409	0.273		
CAL	Baseline	2.76 ± 0.46	2.76 ± 0.37	0.747 (W)	
	4 weeks	2.77 ± 0.47	2.71 ± 0.38	0.688 (W)	
	*p*-value ^a^	0.522	0.021		
	8 weeks	2.73 ± 0.49	2.71 ± 0.42	0.960 (W)	0.165
	*p*-value ^a^	0.270	0.062		
PI	Baseline	0.60 ± 0.34	0.53 ± 0.34	0.243(W)	
	4 weeks	0.50 ± 0.33	0.47 ± 0.26	0.799(W)	
	*p*-value ^a^	0.002	0.156		
	8 weeks	0.46 ± 0.33	0.41 ± 0.27	0.464 (W)	0.699
	*p*-value ^a^	0.000	0.003		

^a^: compared within group; *p*-value for Paired *t*-test. ^b^: compared between groups; *p*-value for two-sample *t*-test (T) or Wilcoxon rank sum test (W). ^c^: compared between groups; *p*-value for repeated-measures ANOVA.

**Table 4 nutrients-16-03000-t004:** Change in clinical parameters by measurement time in the FA set.

Clinical Parameters	Measurement Time	Test (Mean ± SD)	Control (Mean ± SD)	*p*-Value ^b^	*p*-Value ^c^
GI	Baseline	1.69 ± 0.41	1.62 ± 0.41	0.214 (W)	
	4 weeks	1.66 ± 0.49	1.57 ± 0.33	0.097 (W)	
	*p*-value ^a^	0.496	0.175		
	8 weeks	1.58 ± 0.47	1.51 ± 0.44	0.184 (W)	0.926
	*p*-value ^a^	0.016	0.002		
BOP (%)	Baseline	46.83 ± 18.71	41.87 ± 16.52	0.169 (T)	
	4 weeks	43.76 ± 20.63	39.31 ± 13.60	0.230 (W)	
	*p*-value ^a^	0.149	0.118		
	8 weeks	42.23 ± 20.00	37.26 ± 16.99	0.087 (W)	0.972
	*p*-value ^a^	0.035	0.005		
PD	Baseline	2.59 ± 0.35	2.60 ± 0.29	0.712 (W)	
	4 weeks	2.57 ± 0.33	2.54 ± 0.32	0.720 (W)	
	*p*-value ^a^	0.388	0.000		
	8 weeks	2.55 ± 0.33	2.54 ± 0.32	0.945 (W)	0.334
	*p*-value ^a^	0.091	0.003		
GR	Baseline	0.18 ± 0.28	0.17 ± 0.25	0.997 (W)	
	4 weeks	0.21 ± 0.30	0.18 ± 0.26	0.456 (W)	
	*p*-value ^a^	0.002	0.316		
	8 weeks	0.19 ± 0.30	0.18 ± 0.27	0.848 (W)	0.329
	*p*-value ^a^	0.408	0.273		
CAL	Baseline	2.77 ± 0.45	2.77 ± 0.37	0.786 (W)	
	4 weeks	2.78 ± 0.46	2.72 ± 0.38	0.686 (W)	
	*p*-value ^a^	0.588	0.019		
	8 weeks	2.74 ± 0.48	2.71 ± 0.42	0.979 (W)	0.184
	*p*-value ^a^	0.234	0.048		
PI	Baseline	0.60 ± 0.34	0.54 ± 0.34	0.277 (W)	
	4 weeks	0.50 ± 0.32	0.48 ± 0.26	0.777 (W)	
	*p*-value ^a^	0.002	0.134		
	8 weeks	0.47 ± 0.33	0.41 ± 0.27	0.445 (W)	0.732
	*p*-value ^a^	0.000	0.002		

^a^: compared within group; *p*-value for Paired *t*-test. ^b^: compared between groups; *p*-value for two-sample *t*-test (T) or Wilcoxon rank sum test (W). ^c^: compared between groups; *p*-value for repeated-measures ANOVA.

**Table 5 nutrients-16-03000-t005:** Biomarkers in gingival crevicular fluid by measurement time in the PP set.

Biomarkers	Measurement Time	Test (Mean ± SD)	Control (Mean ± SD)	*p*-Value ^b^	*p*-Value ^c^
PGE2 (ng/mL)	Baseline	602.38 ± 178.15	526.36 ± 172.55	0.024 (W)	
	4 weeks	455.80 ± 113.81	518.74 ± 141.58	0.020 (W)	
	*p*-value ^a^	<0.000	0.740		
	8 weeks	345.43 ± 69.02	576.08 ± 186.17	<0.000 (T)	<0.000
	*p*-value ^a^	<0.000	0.079		
IL-1ß (pg/mL)	Baseline	135.45 ± 93.70	114.86 ± 65.96	0.407 (W)	
	4 weeks	102.83 ± 62.54	124.84 ± 71.32	0.109 (W)	
	*p*-value ^a^	0.004	0.240		
	8 weeks	70.89 ± 65.46	141.95 ± 78.02	<0.000 (W)	<0.000
	*p*-value ^a^	<0.000	0.006		
MMP-8 (ng/mL)	Baseline	36.21 ± 18.33	32.74 ± 16.13	0.420 (W)	
	4 weeks	28.30 ± 14.29	34.36 ± 15.41	0.051 (T)	
	*p*-value ^a^	<0.000	0.319		
	8 weeks	21.76 ± 14.65	36.76 ± 16.15	<0.000 (W)	<0.000
	*p*-value ^a^	<0.000	0.039		
MMP-9 (ng/mL)	Baseline	65.19 ± 25.20	60.51 ± 19.81	0.318 (T)	
	4 weeks	49.47 ± 19.97	63.55 ± 20.47	0.001 (T)	
	*p*-value ^a^	<0.000	0.206		
	8 weeks	35.55 ± 20.79	66.46 ± 23.25	<0.000 (W)	<0.000
	*p*-value ^a^	<0.000	0.070		

^a^: compared within group; *p*-value for Paired *t*-test. ^b^: compared between groups; *p*-value for two-sample *t*-test (T) or Wilcoxon rank sum test (W). ^c^: compared between groups; *p*-value for repeated-measures ANOVA.

**Table 6 nutrients-16-03000-t006:** Biomarkers in gingival crevicular fluid by measurement time in the FA set.

Biomarkers	Measurement Time	Test (Mean ± SD)	Control (Mean ± SD)	*p*-Value ^b^	*p*-Value ^c^
PGE2 (ng/mL)	Baseline	590.38 ± 183.91	523.20 ± 172.16	0.049 (W)	
	4 weeks	456.83 ± 111.49	517.22 ± 140.49	0.020 (W)	
	*p*-value ^a^	<0.000	0.791		
	8 weeks	353.76 ± 80.08	575.77 ± 184.23	<0.000 (T)	<0.000
	*p*-value ^a^	<0.000	0.060		
IL-1ß (pg/mL)	Baseline	131.99 ± 93.75	113.65 ± 65.81	0.472 (W)	
	4 weeks	101.19 ± 62.07	124.06 ± 70.79	0.086 (W)	
	*p*-value ^a^	0.005	0.212		
	8 weeks	70.26 ± 64.66	141.07 ± 77.45	<0.000 (W)	<0.000
	*p*-value ^a^	<0.000	0.004		
MMP-8 (ng/mL)	Baseline	35.51 ± 18.58	32.55 ± 16.02	0.401 (T)	
	4 weeks	27.90 ± 14.30	34.31 ± 15.25	0.035 (T)	
	*p*-value ^a^	<0.000	0.271		
	8 weeks	21.69 ± 14.47	36.78 ± 15.98	<0.000 (W)	<0.000
	*p*-value ^a^	<0.000	0.027		
MMP-9 (ng/mL)	Baseline	64.67 ± 26.64	59.71 ± 20.38	0.304 (T)	
	4 weeks	49.49 ± 20.78	62.79 ± 20.94	0.002 (T)	
	*p*-value ^a^	<0.000	0.191		
	8 weeks	36.46 ± 21.54	65.85 ± 23.41	<0.000 (W)	<0.000
	*p*-value ^a^	<0.000	0.057		

^a^: compared within group; *p*-value for Paired *t*-test. ^b^: compared between groups; *p*-value for two-sample *t*-test (T) or Wilcoxon rank sum test (W). ^c^: compared between groups; *p*-value for repeated-measures ANOVA.

**Table 7 nutrients-16-03000-t007:** Safety analysis of control and test at baseline, 8 weeks (mean ± SD).

			Test *n* = 51		Control *n* = 53	*p*-Value ^b^
		*n*	Mean ± SD	*n*	Mean ± SD	
WBC (10^3^/μL)	Baseline	51	6.09 ± 1.42	53	6.07 ± 1.15	0.934 [T]
	8 weeks	47	5.91 ± 1.42	49	6.17 ± 1.58	0.396 [T]
	*p*-value ^a^		0.275		0.962	
RBC (10^6^/μL)	Baseline	51	4.48 ± 0.36	53	4.48 ± 0.39	0.994 [T]
	8 weeks	47	4.45 ± 0.37	49	4.51 ± 0.39	0.282 [T]
	*p*-value ^a^		0.280		0.698	
Hb (g/dL)	Baseline	51	13.57 ± 1.11	53	13.52 ± 1.22	0.828 [T]
	8 weeks	47	13.45 ± 1.25	49	13.61 ± 1.30	0.263 [W]
	*p*-value ^a^		0.257		0.739	
Hct (%)	Baseline	51	40.77 ± 3.29	53	40.64 ± 3.46	0.839 [T]
	8 weeks	47	40.41 ± 3.50	49	40.72 ± 3.50	0.570 [T]
	*p*-value ^a^		0.320		0.772	
Platelet (10^3^/μL)	Baseline	51	250.53 ± 45.59	53	253.45 ± 38.99	0.532 [W]
	8 weeks	47	241.49 ± 45.37	49	249.22 ± 40.50	0.963 [T]
	*p*-value ^a^		0.094		0.153	
ALT (GPT) (IU/L)	Baseline	51	17.80 ± 9.36	53	19.74 ± 12.93	0.646 [W]
	8 weeks	47	17.85 ± 9.91	49	19.24 ± 11.61	0.938 [W]
	*p*-value ^a^		0.791		0.556	
BUN (mg/dL)	Baseline	51	12.99 ± 3.62	53	13.09 ± 3.55	0.878 [W]
	8 weeks	47	13.09 ± 3.70	49	12.58 ± 3.48	0.459 [T]
	*p*-value ^a^		0.768		0.453	
Creatinine (mg/dL)	Baseline	51	0.78 ± 0.17	53	0.77 ± 0.16	0.630 [W]
	8 weeks	47	0.75 ± 0.15	49	0.75 ± 0.15	0.572 [T]
	*p*-value ^a^		0.093		0.118	
Ca (mg/dL)	Baseline	51	9.46 ± 0.34	53	9.52 ± 0.37	0.345 [T]
	8 weeks	47	9.47 ± 0.37	49	9.50 ± 0.30	0.697 [T]
	*p*-value ^a^		0.945		0.620	

^a^: compared within groups; *p*-value for Paired *t*-test. ^b^: compared between groups; *p*-value for two-sample *t*-test [T] or Wilcoxon rank sum test [W].

## Data Availability

The data presented in this study are available on request from the corresponding author due to privacy/ethical restrictions.

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
