# Peer review of "The Clinical Effect of a Propolis and Mangosteen Extract Complex in Subjects with Gingivitis: A Randomized, Double-Blind, and Placebo-Controlled Clinical Trial"

_nutrients, 2024, doi:10.3390/nu16173000_

Round 1

Reviewer 1 Report

Comments and Suggestions for Authors

The study, a multi-centred, randomized, double-blind, placebo-controlled trial, was designed to evaluate the effectiveness and safety of a propolis mangosteen extract complex (PMEC) in improving gingival health in patients with gingivitis and early-stage periodontitis. It involved 104 participants who received either the PMEC or a placebo for eight weeks. The primary objective was to assess changes in inflammatory biomarkers in gingival crevicular fluid (GCF), with clinical parameters like gum inflammation and periodontal health as secondary outcomes. The study revealed that participants in the PMEC group had significantly lower levels of inflammatory biomarkers (IL-1β, PGE2, MMP-8, MMP-9) in their GCF compared to the placebo group after 8 weeks, indicating a reduction in gum inflammation. Although there was a trend toward improvement in clinical parameters in both groups, the differences between the PMEC and placebo groups were not statistically significant. Overall, this well-conducted study with a robust design provides promising findings, particularly regarding biomarker improvements, which could pave the way for further research and potential clinical use of PMEC. It is a well-designed clinical trial. 

I have a few comments/suggestions

Did you consider perhaps going to 3 -6 months? I suggest 8 weeks may be too short to see clinical changes compared to the biomarkers.

Can you add in the methods of how the biomarkers were measured? I am unsure why they are  in percentages, normally reported as a mass e.g. pg/ml

L244, can you add in replicates, mean +- SD or SEM? Please add in all figure legends for the plots

L244 The statistics presented are confusing. Could you find another way to present the data that makes the comparisons clearer for the reader?

L280 makes sure there is a zero before the decimal points in the table – be consistent in all tables.

L386 The study's conclusion is overly optimistic. It emphasizes the significant reduction in inflammatory biomarkers with PMEC supplementation but downplays the lack of corresponding clinical improvements. It suggests PMEC as a promising therapy for periodontal health based on biochemical changes, but this optimism is premature given the absence of significant clinical outcomes. The conclusion should be more balanced by acknowledging the study's limitations, such as the short duration and small sample size, and clearly stating the need for further research to determine the true clinical value of PMEC.

Is there a supplementary information file?

Comments on the Quality of English Language

Please proof read , and check the grammar

Author Response

I uploaded the Point-by-Point responses file

Reviewer 2 Report

Comments and Suggestions for Authors

The authors investigated the effect of propolis and mangosteen extract complex (PMEC) on several biomarkers and clinical parameters of periodontal disease in a randomized, double-blind, placebo-controlled clinical trial. The authors showed that oral administration of PMECs reduced the biomarkers over time, but did not significantly alter clinical parameters. The purpose and experimental method were appropriate and reliable data was obtained. However, there are still several points that need to be improved in the data analysis and presentation.

[Major points]

  1. The authors analyzed longitudinal data in this study with t-test or Wilcoxon rank-sum test (comparison between Test and Control groups) or paired t-test (comparison between baseline and 4, 8 weeks). Indeed, this type of analysis is commonly used. However, repeated-measures 2-way ANOVA or linear mixed models are preferred to investigate whether changes in clinical parameters and biomarkers over time depends on PMEC treatment (the latter method is preferable). In detail, the explanatory variables are (1) treatment, (2) time, and (3) the interaction term between treatment and time.

  2. Paired t-test is performed in FA set and safety set (Tables 4, 6 and 7). However, paired t-test is used only when data are collected twice from the same subject (i.e. sample sizes are the same). Authors should specify how they handled missing data.

[Minor points]

  1. Please specify the type of randomization (simple randomization, blocked randomization, or stratified randomization) in Methods section (page 3, line 103).

  2. The authors showed the results of the analysis from both the FA set (ITT protocol) and the PP set. The authors should describe in the discussion section which of these two results they consider more important.

  3. The description about subjects (Page 5, line 191-198) are difficult to understand because the words used are different from that in Figure 1. 

  4. Three significant digits for P values are sufficient in Abstract (line 18) and in Tables 2 to 7. Also, one decimal place is sufficient in weight (kg) and height (cm) of Table 2.

  5. The meaning of symbols (T, C and W) should be described in Table 2.

  6. (typo) "Control" in the top right cell of Table 3 is "P-value".

Author Response

(The authors gave the same response as above.)

Round 2

Reviewer 2 Report

Comments and Suggestions for Authors

The authors answered and corrected the manuscript in response to all comments.

I understood the handling method for missing data in this study.

Please add the paragraph for handling missing data (about LOCF method) to the Methods section.

Author Response

The LOCF method was added in Method section (L 193, Page 5): In a two-sample t-test, Last Observation Carried Forward (LOCF) method was utilized to address missing data within the FA set. Specially, for missing data at the 8-week time point, the corresponding 4-week data were imputed for analysis. However, missing data at the 4-week time point were not substituted with baseline measurements.
